# Nutritional status and associated factors among adult patients with tuberculosis in public hospitals of Sidama Region, Ethiopia

**Amelo Bolka**[1]*, **Selamawit Sikuare**[1], **Assefa Philipos Kare**[1], **Fentaw Wassie Feleke**[2,3], **Tafese Bosha**[2]

**1** School of Public Health, Yirgalem Hospital Medical College, Yirgalem, Ethiopia, **2** School of Nutrition, Food Science and Technology, Hawassa University, Hawassa, Ethiopia, **3** Department of Public Health, School of Public Health, Woldia University, Woldia, Ethiopia

* amelobolka@gmail.com

## Abstract

In Ethiopia, while undernutrition among adult patients with tuberculosis (PWTB) is well-documented, evidence on anemia and its coexistence with undernutrition remains limited. This study was aimed at assessing nutritional status and associated factors among adult PWTB attending public hospitals of Sidama Region. A facility-based cross-sectional study was conducted from 4 February to 27 March 2024 among 418 systematically sampled PWTB from public hospitals. Data were collected using pretested structured questionnaires and analyzed in STATA 14. Modified Poisson regression with robust standard errors identified factors associated with nutritional status, presented as adjusted prevalence ratios (APR) with 95% confidence intervals (CI). The magnitudes of undernutrition and anemia were 37.1% (95% CI: 32.4–41.7) and 48.6% (95% CI: 43.7–53.4) respectively. Intestinal parasites were detected in 25.1% (95% CI: 20.9–29.3) of adult PWTB. Anemia prevalence was significantly higher in undernourished PWTB (87.1%) compared to those with normal nutrition (25.8%; p<0.001). Undernutrition was associated with food-insecurity (APR=1.89; 95% CI: 1.47–2.45), low dietary diversity (APR=1.56; 95% CI: 1.21–2.01), TB-HIV coinfection (APR=1.65; 95% CI: 1.23–2.21), and parasite infection (APR=1.78; 95% CI: 1.41–2.25). The identified predictors of anemia among PWTB were food insecurity (APR=1.41; 95% CI: 1.16–1.73), parasitic infection (APR=1.23; 95% CI: 1.01–1.51), and TB-HIV coinfected (APR=1.36; 95% CI: 1.04–1.77). The study revealed a prevalence of undernutrition and anemia among PWTB, with key predictors including poor dietary diversity, food insecurity, TB-HIV coinfection, and parasitic infection. Given the public health significance of anemia, especially among undernourished patients, we recommend integrating routine nutritional screening and targeted interventions—such as food support, parasite control, and HIV care—into TB treatment programs.

**Data availability statement:** All relevant data are within the manuscript and its Supporting Information files.

**Funding:** The author(s) received no specific funding for this work.

**Competing interests:** The authors have declared that no competing interests exist.

## Background

Tuberculosis continues to pose a formidable challenge to global public health systems, maintaining its position as one of the most devastating infectious diseases affecting humanity [1]. Current epidemiological data reveals TB remains the deadliest infectious disease pre-and post-COVID, with mortality rates exceeding all other pathogens [2]. According to the World Health Organization (WHO) of 2024, an estimated 10.8 million people fell ill with TB in 2023, with the majority residing in low- and middle-income countries [3]. This disease distribution exhibits marked geographical inequality, with low-and middle-income countries (LMICs) bearing the brunt of the epidemic due to systemic health disparities. TB and malnutrition interact synergistically in vulnerable groups, worsening disease outcomes [4–6].

Undernutrition constitutes a prevalent but frequently neglected complication in individuals undergoing treatment for tuberculosis, exerting a substantial impact on disease trajectory and therapeutic success [7]. Deficits in macronutrients and essential micronutrients, along with progressive loss of lean body mass, are frequently observed in this population, contributing to compromised immunological defences and hindered convalescence [8]. Undernourished individuals face a threefold higher risk of developing active TB than those with proper nutrition, per empirical evidence [9]. Moreover, nutritional insufficiency in PWTB correlates with increased fatality rates, extended treatment courses, and a heightened susceptibility to adverse drug reactions [10].

Anaemia is another common comorbidity among PWTB and is frequently observed in conjunction with undernutrition [11]. Its underlying mechanisms are diverse and complex, typically involving sustained inflammatory processes, deficits in key micronutrients—most notably iron, folate, and vitamin B12—and diminished haematopoietic activity attributable to the systemic impact of infection. This haematological impairment not only exacerbates fatigue and limits physical capacity but may also hinder immunological efficiency and compromise adherence to antitubercular therapy [12]. As such, anaemia constitutes a clinically relevant factor that warrants systematic evaluation within the broader framework of TB management [11].

Multiple intersecting factors influence malnutrition patterns in TB populations [13]. Economic deprivation, household food shortages, concurrent HIV infection, advanced disease severity, and medication-related complications collectively exacerbate nutritional deficits [14,15]. The complex interplay between these variables - ranging from structural inequities to biological factors - requires careful examination to identify priority intervention points [16]. Disentangling their individual and synergistic impacts enables development of stratified nutritional support strategies tailored to patients' specific risk profiles [17]. Such precision approaches could optimize treatment outcomes by addressing the causes of wasting and micronutrient depletion during anti-tuberculosis therapy [18].

Recognising the interplay between nutrition and TB, the WHO and national TB control programmes advocate for nutritional assessment and support as part of standard care [19,20]. However, implementation remains inconsistent, and empirical

evidence on the effectiveness of nutritional interventions in improving TB outcomes—particularly in high-burden settings like Ethiopia—is lacking [21]. Context-specific data are urgently needed to guide tailored nutrition-TB integration policies and their practical implementation [18]. This highlights the importance of early nutritional screening and intervention as part of standard TB care [22].

Ethiopia's National Tuberculosis Program delivers TB treatment through public and private health facilities as part of its public health strategy for adults [23]. Studies have assessed undernutrition among adult PWTB attending these treatment facilities [24–27]. However, there is insufficient information regarding anemia and the co-existence of undernutrition and anemia in this population. Moreover, the regional prevalence of undernutrition and anemia has yet not been assessed. This gap highlights the importance of estimating the prevalence of undernutrition and anemia and identifying the factors associated with them. Therefore, this study aimed to assess nutritional status and associated factors among adult PWTB attending public hospitals in the Sidama region of Ethiopia.

## Methods and materials

### Ethical approval

This study was conducted in accordance with the Declaration of Helsinki. Ethical approval was obtained from the Institutional Review Board of Yirgalem Hospital Medical College (Protocol Number: YHMC/IRB005, Date: 2/11/2023). Written informed consent was obtained from all study participants. Nutritional counseling was provided to participants identified with undernutrition or anemia. Participant information was kept confidential using pseudonymous codes.

### Study setting

This study was conducted in public hospitals in Sidama Region, southern Ethiopia. The region is located 273 kilometers south of Addis Ababa, the nation's capital. According to the Central Statistics Agency of Ethiopia's report, the region's total population was estimated to be 4,748,623. Based on the 2024 Regional Health Bureau Report, there are 23 public hospitals, 137 health centers, 551 health posts, 6 specialty centers, 35 specialty clinics, 247 clinics (144 medium, 96 primary, and 7 institutions based), 4 diagnostic laboratories, 389 drug stores, and 82 pharmacies providing health services to the region's population. All public hospitals and health centers, along with selected private health institutions, provide TB detection and treatment services [28].

### Study design and period

We conducted a facility-based cross-sectional study from 4 February to 27 March 2024 in public hospitals in the Sidama Region, involving 418 adult PWTB.

### Study population and eligibility criteria

The study population consisted of adult PWTB undergoing treatment follow-up in the aforementioned hospitals. Adult PWTB who were pregnant or lactating; had known chronic conditions (such as diabetes, chronic kidney or liver disease, or cancer); had drug-resistant TB; were enrolled in nutritional support programs; or were critically ill or physically unable to undergo anthropometric measurements were excluded.

### Sample size determination and sampling procedures

The sample size for assessing nutritional status was calculated using the single population proportion formula, with the following parameters: a 95% confidence level, a 5% margin of error, and a 10% non-response rate. Based on a previously reported undernutrition prevalence of 43.6% in a similar population [29], the sample size was determined to be 418. For anemia, the sample size was also estimated using the single population proportion formula, with the same confidence

level, margin of error, and non-response rate. Anemia prevalence among tuberculosis of 69% was used from a previous study [11] yielding a sample size of 362. The larger sample size of 418 was adopted for the study.

We used a simple random sampling method to select ten hospitals (*Adare, Bona, Bursa, Daye, Hula, Kawado, Leku, Tula, Wondo Genet, and Yirgalem*) out of twenty-three. A systematic random sampling method was then used to select participants from each selected hospital. The sample size was proportionally allocated to the selected hospitals based on the number of patients attending TB clinics. All PWTB who fulfilled the inclusion criteria were enrolled.

## Study variables

The dependent variables of interest were anemia and undernutrition. Undernutrition was diagnosed in tuberculosis (TB) patients with a BMI below 18.5 kg/m$^2$; those with a BMI ≥ 18.5 kg/m$^2$ were classified as normally nourished. Undernutrition severity was categorized as mild (BMI 17.0–18.4 kg/m$^2$), moderate (BMI 16.0–16.9 kg/m$^2$), and severe (BMI < 16.0 kg/m$^2$) [30]. Anemia was defined as a hemoglobin level <12 g/dL for women and <13 g/dL for men among PWTB [31].

We considered the following independent variables for multivariable analysis: sex, family size, educational status, wealth index, food security, dietary diversity, meal frequency, enrolment in a safety-net program, nutrition education, nutritional knowledge, nutritional care and support, TB treatment phase, TB-HIV coinfection, parasite infection, and eating problems.

## Data collection instruments and procedures

Trained data collectors collected data using a structured, interviewer-administered questionnaire adapted from relevant literature [3–5]. The questionnaire covered socio-demographic characteristics, environmental factors, dietary factors, life-style factors, and clinical factors. To ensure privacy, we recruited data collectors from among the TB treatment providers at each TB clinic.

Dietary diversity of participants was measured using the standard Food and Agriculture Organization of the United Nations (FAO) tool, assessing food consumption over the previous 24 hours. All foods eaten by participants - whether consumed inside or outside the home were included. The individual dietary diversity scale categorized foods into nine groups: starchy staples; vitamin-A-rich fruits and vegetables; other fruits and vegetables; meats and fish; dark green leafy vegetables; organ meat; eggs; legumes, nuts, and seeds; and milk and milk products. PWTB who consumed five or more out of nine food groups were considered to have a high dietary diversity score [32].

Household food security was measured using the standard Household Food Insecurity Access Scale (HFIAS). Adult PWTB were asked nine standard questions about difficulties their families had faced in the previous 28 days. Based on the standard scale, household food insecurity was classified into two categories: food secure and food insecure [33].

Study participants' nutritional knowledge was assessed using sixteen questions. The questions aimed to determine whether patients know about nutrition, the causes, consequences, and prevention methods of undernutrition. Scores were assigned to each response, with correct answers receiving one point and incorrect answers receiving zero. Study participants who correctly answered thirteen or more of the sixteen (≥80%) knowledge questions were considered as having good knowledge whereas below thirteen were poor knowledge.

Weight and height measurements were taken following standard anthropometric procedures. Weight was measured using a SECA digital scale (Seca GmbH & Co. KG) to the nearest 0.1 kg, with subjects wearing light clothing and no footwear. Height was measured using a portable stadiometer to the nearest 0.1 cm, with subjects barefoot, heels together, head positioned in the Frankfurt plane against the stadiometer, and eyes looking straight ahead. Height and weight were measured twice and average measurements used for BIM calculation.

Capillary blood samples were collected from each respondent using a finger-prick method. Following standard procedures, the middle finger of the left hand was cleaned and pricked. The first blood drop was cleaned off, and the second

drop was collected into a micro cuvette for hemoglobin measurement using the HemoCue 301 system. Hemoglobin values were adjusted for altitude based on WHO guidelines [31].

A microscopic stool examination was conducted to detect parasitic infections. Fresh stool samples were obtained from participants, with PWTB instructed on proper collection procedures using clean, pre-labelled containers. The formal-ether concentration technique was applied to detect parasites (eggs, cysts or oocysts). Direct microscopic observation method was followed utilizing concentration method to identify the presence of intestinal parasites effectively.

### Quality assurance

Qualified data collectors and supervisors received comprehensive training on the Kobo Toolbox system and interview skills. The data collection tool was pre-tested on 5% of the sample, and necessary modifications were made. Anthropometric measurements were taken using calibrated scales. Rigorous supervision included daily checks and prompt error corrections. The use of the Kobo Toolbox system for data collection facilitated logical data entry and maintained data quality. Investigators verified form completeness, maintaining data integrity throughout.

### Data management and analysis

Data were collected using the Kobo Toolbox system and exported and analysed with STATA version 14. Body mass index was calculated as weight in kilograms divided by the square of height in meters ($kg/m^2$). The household's wealth index was calculated using principal component analysis (PCA) based on the ownership of valuable assets, housing conditions and access to social services. Data were described using frequency distributions, measures of central tendency and dispersion.

Associated factors of nutritional status were determined using modified Poisson regression with robust standard errors. We used modified Poisson regression with robust standard errors instead of logistic regression, as odds ratios from logistic regression can overestimate risk for high-prevalence outcomes [34–36]. In addition, compared to the odds ratio, the prevalence ratio is simpler to explain and understand for general audiences [35]. The best-fitting model was selected using Akaike's Information Criterion (AIC), Bayesian Information Criterion (BIC), and log-likelihood with likelihood ratio tests. The model with the lowest AIC/BIC values and a statistically significant likelihood ratio test was chosen [37]. We evaluated multicollinearity among independent variables using multivariable linear regression, with a variance inflation factor < 5 indicating low multicollinearity for all variables [38]. Multivariable analyses were performed to control for confounders. Variables with a $p$-value < 0.25 in the bivariable analysis were included as candidates for the multivariable regression model. Statistical significance was set at $p < 0.05$. We presented significant associations using adjusted prevalence ratios (APRs) with 95% confidence intervals.

## Results

### Socio-demographic characteristics of adult PWTB

This study included 418 PWTB, resulting in a response rate of 100%. The median age (interquartile range) of the participants was 34 (IQR: 25, 41) years. Slightly more than half (55.7%) of the respondents were female. One-third (33.7%) of the respondents were Sidama in ethnicity. Slightly less than two-thirds (63.6%) of the study participants were married. Two hundred sixty-three (62.9%) of the respondents identified as followers of the Protestant religion. Two-thirds (67.7%) of the study participants had attended primary education or above. The majority (80.4%) of the PWTB resided in urban areas. More than half (59.8%) of the respondents had a family size of greater than or equal to five. Seventy-five (17.9%) of the study participants or their households were enrolled in productive safetynet program. Pertaining to wealth tertiles, slightly less than half (47.4%) of the study participants households were categorized as having a lower wealth index (Table 1).

**Table 1. Socio-demographic characteristics of adult PWTB attended hospitals of Sidama Region.**

| Variable (n = 418) | Category | Frequency | Percent (%) |
|---|---|---|---|
| Age | < 34 years | 207 | 49.5 |
| | ≥ 34 years | 211 | 50.5 |
| Residence | Urban | 336 | 80.4 |
| | Rural | 82 | 19.6 |
| Sex | Male | 185 | 44.3 |
| | Female | 233 | 55.7 |
| Marital status | Married | 266 | 63.6 |
| | Single | 88 | 21.1 |
| | Widowed | 64 | 15.3 |
| Religion | Protestant | 263 | 62.9 |
| | Orthodox | 106 | 25.4 |
| | Muslim | 49 | 11.7 |
| Ethnicity | Sidama | 141 | 33.7 |
| | Amhara | 89 | 21.1 |
| | Oromo | 79 | 18.9 |
| | Gurage | 55 | 13.2 |
| | Wolaita | 54 | 12.9 |
| Education level | No formal education | 135 | 32.3 |
| | Attended primary education | 283 | 67.7 |
| Family size | ≤5 | 168 | 50.2 |
| | >5 | 250 | 59.8 |
| Enrolled in safetynet program | Yes | 75 | 17.9 |
| | No | 343 | 82.1 |
| Wealth index | Lower | 189 | 47.4 |
| | Medium | 126 | 30.1 |
| | Higher | 94 | 22.4 |

### Household food security, nutritional knowledge, and dietary diversity among adult PWTB

Based on the HFIAS scale, nearly half (47.8%) of PWTB were from food-insecure households. Among study participants, 68.7% had poor nutrition knowledge, and seven in ten (69.4%) did not receive nutritional care/support during TB treatment (Table 2)

Two-thirds (67.1%) of adult PWTB attending public hospitals consumed ≤ 2 meals the previous day. The mean ± SD dietary diversity score of the study participants was 4.5 ± 1.2. One hundred seventy-eight (42.6%) of the study participants had consumed fewer than five food groups (less diversified diet) the previous day (Table 2).

### Clinical factors among adult PWTB

The majority of the PWTB (86.4%) were newly diagnosed. Four-fifths of the study participants (80.4%) had a positive smear result. Seventy-three (17.5%) of the study participants had a family history of TB. Thirty-three (7.9%) study partici-pants reported TB-HIV co-infection. Ninety-nine (23.7%) of the PWTB reported experiencing some type of eating problem. Regarding the duration (phase) of TB treatment, slightly more than half (54.1%) reported being in the continuation phase (≥ 2 months). Fifty-six patients (13.4%) were undergoing retreatment (Table 3).

**Global Public Health**

**Table 2. Dietary and nutritional characteristics of adult PWTB attended public hospitals of Sidama Region.**

| Variables (n = 418) | Category | Frequency | Percent (%) |
|---|---|---|---|
| Household food security status | Secure | 229 | 52.2 |
| | Insecure | 189 | 47.8 |
| Nutritional knowledge | Poor | 287 | 68.7 |
| | Good | 131 | 31.3 |
| Nutritional care and support | No | 290 | 69.4 |
| | Yes | 128 | 30.6 |
| Number of meals consumed in previous day | ≥ 3meals | 137 | 32.8 |
| | ≤ 2 meals | 281 | 67.1 |
| Dietary diversity | Low | 178 | 42.6 |
| | High | 240 | 57.4 |

**Table 3. Clinical factors of adult PWTB attended public hospitals of Sidama Region.**

| Variables (n = 418) | Categories | Frequency | Percent (%) |
|---|---|---|---|
| Type of TB treatment | New | 362 | 86.6 |
| | Retreatment | 56 | 13.4 |
| Smear result | Positive | 336 | 80.4 |
| | Negative | 82 | 19.6 |
| Family history of TB | Yes | 73 | 17.5 |
| | No | 345 | 82.5 |
| HIV co-infection | Yes | 33 | 7.9 |
| | No | 385 | 92.1 |
| Treatment duration (phase) | Intensive | 192 | 45.9 |
| | Continuation | 226 | 54.1 |
| Eating problem | No | 99 | 76.3 |
| | Yes | 319 | 23.7 |

## Environmental related factors

The vast majority (93.8%) of study participants' families owned a latrine. Four-fifths (80.6%) of these latrines were pit latrines. Nearly a quarter (23.7%) reported having a handwashing facility near the toilet. Slightly more than one-fourth (26.3%) of study participants reported washing their hands with soap or ash after toilet use. Eight in ten (82.5%) obtained drinking water from protected sources (Table 4).

## Parasite infection among adult PWTB

Stool examination identified eight species of intestinal parasites. Among adult PWTB, one-fourth (25.1%; 95% CI: 20.9–29.3) were infected with at least one intestinal parasite. Nineteen adult PWTB (4.5%) were infected with two or more parasites. The most prevalent parasite was *Ascaris lumbricoides* (*A.lumbricoides*) (5.7%), followed by *Giardia lamblia* (*G.lamblia*) (4.5%), while *Schistosoma mansoni* (*S.mansoni*) was the least prevalent (1.2%) (Fig 1).

## Anemia among adult PWTB

The mean (±SD) hemoglobin level among PWTB was 12.73 ± 1.75 g/dL. The prevalence of anemia was 48.6% (95% CI: 43.7–53.4). Among anemic PWTB, 13.6% had moderate anemia and 34.9% had mild anemia. Two-thirds (66.5%) of

**Table 4. Environmental related factors among adult PWTB attended public hospitals of Sidama Region.**

| Variables (n = 418) | Categories | Frequency | Percent (%) |
|---|---|---|---|
| Own latrine | Yes | 392 | 93.8 |
| | No | 26 | 6.2 |
| Type of latrine (n = 392) | Pit latrine | 316 | 80.6 |
| | Other type | 76 | 19.4 |
| Hand washing facility | Yes | 99 | 23.7 |
| | No | 319 | 76.3 |
| Hand wash with soap | Yes | 110 | 26.3 |
| | No | 308 | 73.7 |
| Source of drinking water | Protected | 345 | 82.5 |
| | Unprotected | 73 | 17.5 |

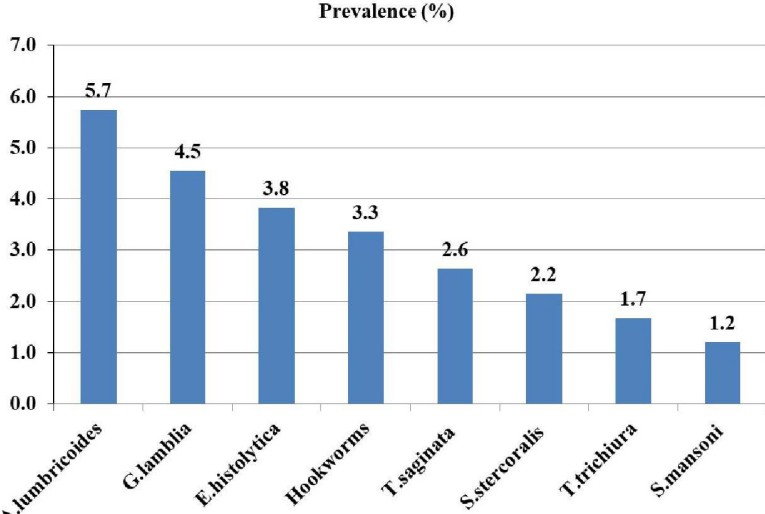

**Fig 1. Parasite infection among adult TB patients attended public hospitals of Sidama Reigion.**

anemic PWTB were undernourished. Anemia was significantly more prevalent among undernourished PWTB (87.1%) than normally nourished patients (25.8%; p < 0.001). More than half (53.3%) of the parasite infected PWTB were anemic (Fig 2).

## Undernutrition among adult PWTB

The prevalence of undernutrition among PWTB was 37.1% (95% CI: 32.4–41.7). The proportions of severe, moderate, and mild undernutrition were 5.5%, 12.0%, and 19.6%, respectively. The vast majority (87.1%) of undernourished PWTB were anemic. Undernutrition was more prevalent among females (43.2%) than males (30.2%; p = 0.010). More than half (59%) of the parasite-infected PWTB were undernourished (Fig 3).

## Predictors of undernutrition and anemia among PWTB

Among potential predictors of undernutrition, ten variables met the criteria for inclusion in the multivariable analysis: educational level, nutritional knowledge, nutritional care and support, food security, dietary diversity, meal frequency, eating

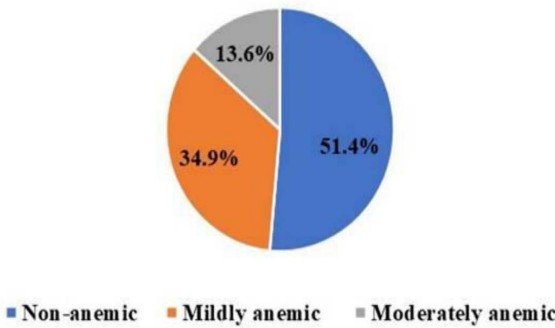

**Fig 2. Anemia among adult TB patients attended public hospitals of Sidama Reigion.**

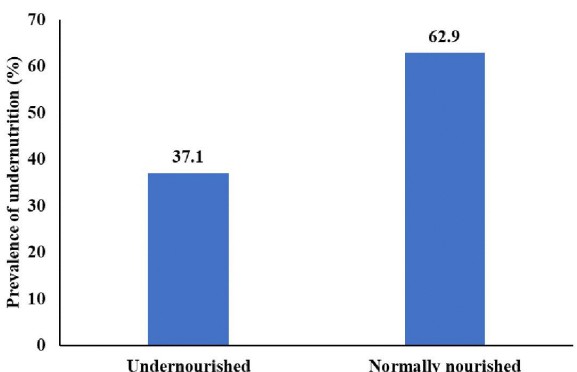

**Fig 3. Under nutrition among adult TB patients attended public hospitals of Sidama Reigion.**

problems, TB treatment phase, TB-HIV coinfection, and parasite infection. Modified Poisson regression analysis identified four significant predictors of undernutrition: food security, dietary diversity, TB-HIV co-infection, and parasite infection (Table 5).

Patients with tuberculosis from food-insecure households had an 89% higher prevalence of undernutrition (APR = 1.89; 95% CI: 1.47–2.45) than those from food-secure households. Those with inadequate dietary diversity exhibited a 56% higher prevalence (APR = 1.56; 95% CI: 1.21–2.01) compared to those with adequate dietary diversity. TB-HIV co-infected patients had 65% higher likelihood of undernutrition (APR = 1.65; 95% CI: 1.23–2.21) relative to TB-only patients. Similarly, intestinal parasite infection increased the likelihood by 78% (APR = 1.78; 95% CI: 1.41–2.25) compared to no infection (Table 5).

Among potential predictors of anemia, the following variables met the inclusion criteria and were entered into the modified multivariable Poisson regression model: sex, family size, educational status, wealth index, food security, enrollment in a safety-net program, nutrition education, TB-HIV coinfection, parasite infection, and eating problems. In the final model, food security, parasite infection, and TB-HIV coinfection were significant predictors of anemia.

Patients with tuberculosis from food-insecure households had a 41% higher likelihood of developing anemia than their food-secure counterparts (APR = 1.41; 95% CI: 1.16–1.73). Those with parasitic infection had a 23% higher likelihood (APR = 1.23; 95% CI: 1.01–1.51), while TB-HIV coinfected patients had a 36% higher likelihood (APR = 1.36; 95% CI: 1.04–1.77) of developing anemia compared to their respective counterparts (Table 6).

**Table 5. A bivariable and multivariable modified Poisson regression analysis of predictor of nutritional status among PWTB attended public hospitals of Sidama Region.**

| Variable (n = 418) | Nutritional status | | CPR | APR (95% CI) | P-value |
|---|---|---|---|---|---|
| | Undernourished | N. nourished | | | |
| Educational status | | | | | |
| No formal education | 49 (36.3%) | 86 (63.7%) | 1.03 | 1.01 (0.78, 1.31) | 0.929 |
| Primary education | 106 (34.5%) | 177 (62.5%) | 1.00 | 1.00 | |
| Nutritional knowledge | | | | | |
| Poor | 97 (33.8%) | 190 (66.2%) | 1.31 | 1.23 (0.97, 1.56) | 0.089 |
| Good | 58 (44.3%) | 73 (55.7) | 1.00 | 1.00 | |
| Nutritional care | | | | | |
| No | 108 (37.2%) | 182 (62.8%) | 0.99 | 1.01 (0.79, 1.30) | 0.917 |
| Yes | 47 (36.7%) | 81 (63.3%) | 1.00 | 1.00 | |
| Dietary diversity | | | | | |
| Low | 89 (50%) | 89 (50%) | 1.82 | 1.56 (1.21, 2.01) | 0.001* |
| High | 66 (27.5%) | 174 (72.5%) | 1.00 | 1.00 | |
| Meal frequency | | | | | |
| ≤ 2 meals | 105 (37.4%) | 176 (62.6%) | 0.97 | 0.94 (0.73, 1.21) | 0.639 |
| ≥ 3meals | 49 (36%) | 87 (64%) | 1.00 | 1.00 | |
| Treatment phase | | | | | |
| Intensive | 72 (37.5%) | 120 (62.5%) | 0.98 | 0.96 (0.76, 1.22) | 0.765 |
| Continuation | 83 (36.7%) | 143 (63.3%) | 1.00 | 1.00 | |
| TB-HIV-coinfection | | | | | |
| Yes | 21 (63.6%) | 12 (36.4%) | 1.83 | 1.65 (1.23, 2.21) | 0.001* |
| No | 134 (34.8%) | 251(65.2%) | 1.00 | 1.00 | |
| Parasite infection | | | | | |
| Yes | 62 (59% | 43 (41%) | 1.99 | 1.78 (1.41, 2.25) | < 0.001* |
| No | 93 (29.7%) | 220 (70.3%) | 1.00 | 1.00 | |
| Eating problem | | | | | |
| Yes | 45 (45.5%) | 54 (54.5%) | 1.32 | 1.10 (0.84, 1.44) | 0.486 |
| No | 110 (34.5%) | 209 (65.5%) | 1.00 | 1.00 | |
| Food security | | | | | |
| Insecure | 100 (50%) | 100 (50%) | 1.98 | 1.89 (1.47, 2.45) | < 0.001* |
| Secure | 55 (25.2%) | 163 (74.8%) | 1.00 | 1.00 | |

APR: Adjusted Prevalence Ratio; CI: Confidence Interval; CPR: Crude Prevalence Ratio; HIV: Human Immune Virus; *Significant association (P < 0.05); TB: Tuberculosis; 1.00: Reference Group

## Discussion

A facility-based cross-sectional study was conducted to assess undernutrition and associated factors among PWTB in Sidama Region. The study found that 37.1% of the participants were undernourished. This study found that 48.6% of PWTB presented anemia. The majority (87.1%) of undernourished PWTB were anemic. Predictors of undernutrition included household food security status, dietary diversity score, intestinal parasite infection, and TB-HIV co-infection. Predictors of anemia were food security status, parasite infection and TB-HIV co-infection.

Our study presented the 37.1% prevalence of undernutrition among PWTB attended public hospitals of Sidama Region. Studies from southern Ethiopia (38.9%) [24] and Addis Ababa (39.7%) [25] presented consistent findings. However

**Table 6. A bivariable and multivariable modified Poisson regression analysis of predictor of anemia among PWTB attended public hospitals of Sidama Region.**

| Variable (n = 418) | Anemia | | CPR | APR (95% CI) | P-value |
|---|---|---|---|---|---|
| | **Anemic** | **Non-anemic** | | | |
| Sex | | | | | |
| Female | 112 (48%) | 121 (52%) | 1.02 | 1.04 (0.86, 1.26) | 0.662 |
| Male | 91 (49.2%) | 94 (50.8%) | 1.00 | 1.00 | |
| Educational status | | | | | |
| No formal education | 61 (45.2%) | 74 (54.8%) | 1.00 | 1.00 | |
| Primary education | 142 (50.2%) | 141 (49.8%) | 1.11 | 1.13 (0.91, 1.40) | 0.270 |
| Family size | | | | | |
| ≥ 5 members | 120 (48%) | 130 (52%) | 1.26 | 1.03 (0.85, 1.25) | 0.765 |
| < 5 members | 83 (49.4%) | 85 (50.6%) | 1.00 | 1.00 | |
| Nutritional education | | | | | |
| No | 122 (49.4%) | 125 (50.6%) | 1.14 | 1.92 (0.76, 1.12) | 0.438 |
| Yes | 81(47.4%) | 90 (52.6%) | 1.00 | 1.00 | |
| TB-HIV-coinfection | | | | | |
| Yes | 22 (66.7%) | 11 (33.3%) | 1.41 | 1.36 (1.04, 1.77) | 0.023* |
| No | 181(47%) | 204 (53%) | 1.00 | 1.00 | |
| Parasite infection | | | | | |
| Yes | 61 (58.1%) | 44 (41.9%) | 1.28 | 1.23 (1.01, 1.51) | 0.038* |
| No | 142 (45.4%) | 171 (54.6%) | 1.00 | 1.00 | |
| Eating problem | | | | | |
| Yes | 54 (54.5%) | 45 (45.5%) | 1.17 | 1.13 (0.91, 1.41) | 0.247 |
| No | 149 (46.7%) | 170 (53.3%) | 1.00 | 1.00 | |
| Food security | | | | | |
| Insecure | 116 (58%) | 84 (42%) | 1.45 | 1.41 (1.16, 1.73) | 0.001* |
| Secure | 87 (40%) | 131 (60%) | 1.00 | 1.00 | |
| Wealth index | | | | | |
| Poor | 85 (42.9%) | 113 (57.1%) | 1.26 | 1.24 (0.98, 1.57) | 0.076 |
| Middle | 68 (54%) | 58 (46%) | 1.24 | 1.21 (0.95, 1.54) | 0.120 |
| Rich | 50 (53.2%) | 44 (46.8%) | 1.00 | 1.00 | |
| Safetynet enrolment | | | | | |
| Yes | 40 (53.3%) | 35 (46.7%) | 1.00 | 1.00 | 0.762 |
| No | 163 (47.5%) | 180 (52.5%) | 1.12 | 1.03 (0.82, 1.31) | |

APR: Adjusted Prevalence Ratio; CI: Confidence Interval; CPR: Crude Prevalence Ratio; HIV: Human Immune Virus; *Significant association (P < 0.05); TB: Tuberculosis; 1.00: Reference Group.

studies conducted in Haromaya (43.6%) [29], southwest Ethiopia (43.9%) [26], east Ethiopia (44.3%) [27], and northwest Ethiopia (57.1%) [40] reported higher prevalence of undernutrition among adult PWTB. The discrepancies in prevalence rates could be due to differences in socioeconomic status, and comorbidities (HIV, or parasitic infections). Moreover, food insecurity, variations in dietary habits, cultural dietary practices, adherence to TB treatment, and disease severity at diagnosis further contribute to these disparities. Some studies measured the outcome of interest (BMI) at the time of TB diagnosis, while others measured it at the end of treatment after patients had recovered—this difference in timing could also contribute to the observed variations.

This study found that 48.6% of PWTB presented anemia, indicating severe public health significance in the study population. Moreover, nine in ten (87.1%) undernourished PWTB were anemic. Comparable finding was reported from eastern Sudan (44%) [41], while higher prevalences were observed in Jimma, Ethiopia (55%) [39], Ghana (69%) [42], and Tanzania (86%) [43], and a systematic review of African adult PWTB (69%) [44]. The high anemia prevalence in PWTB could stems from multiple factors: increased metabolic demand exacerbating iron deficiency [10], chronic inflammation elevating hepcidin (which impairs iron absorption and recycling) [45], comorbidities (parasitic infections) depleting nutrients [46], and reduced dietary intake due to TB-associated anorexia and gastrointestinal disturbances [47].

We found that household food security was significantly associated with undernutrition and anemia among PWTB in the study area. Food insecurity increased the likelihood of undernutrition by 89% and anemia by 41% compared to PWTB with food secured households. Studies have so far identified food insecurity as an important predictor of nutritional status of population of interest [48,49]. This could be due to the fact that household food insecurity impacted the availability, accessibility, and utilization of nutrient-rich foods among PWTB impairing their ability to meet heightened metabolic demands [48].

We found that dietary diversity score was significantly associated with undernutrition and anemia among PWTB in the study area. PWTB with a low dietary diversity score exhibited 56% higher prevalence of undernutrition compared to those with an adequate dietary diversity score. This finding was in line with findings from eastern Ethiopia [29]. Inadequate dietary diversity contributes to undernutrition in PWTB by failing to meet their elevated energy and immune needs while reducing micronutrient adequacy, compounding nutritional deficits.

The study also presented that TB-HIV co-infection was significantly associated with undernutrition and anemia among the study population. TB-HIV co-infected patients had 65% higher likelihood of undernutrition and 36% higher likelihood of anemia compared to PWTB without HIV co-infection. This finding aligns with studies conducted in southern Ethiopia [24], and southwest Ethiopia [26]. This could be due to the fact that TB-HIV co-infection exacerbates metabolic demands, chronic inflammation, malabsorption, and immune dysfunction, while also impairing nutrient utilization and appetite [10,45].

The study witnessed a significant association between undernutrition and parasite infection among PWTB in the study area. Parasite infection increased the likelihood of undernutrition by 79% and anemia by 23% compared to PWTB without intestinal parasites. This aligns with studies in east Shoa, Ethiopia [50] and eastern Ethiopia [51] that linked parasitic infections to undernutrition in PWTB. Mechanistically, these infections result in undernutrition through nutrient malabsorption, chronic blood loss, and systemic inflammation, while concurrently elevating metabolic demands [52].

BMI alone may underestimate the true burden of nutritional deficiency, as it primarily reflects overall body weight relative to height, without accounting for the nuances of dietary quality and micronutrient intake [53]. Many individuals classified as having a normal BMI may still grapple with unmeasured micronutrient deficiencies, especially in contexts characterized by low dietary diversity and inadequate access to a variety of nutrient-rich foods. This phenomenon, often referred to as hidden hunger, presents a significant challenge in assessing the nutritional status of adults with tuberculosis [54]. A comprehensive nutritional assessment that includes both macronutrient and micronutrient evaluation is essential to fully understand and address the nutritional needs of PWTB, ultimately improving their clinical management and recovery prospects [55].

The findings of this study should be interpreted in light of the following limitations. Given that this study followed a cross-sectional design, it was not possible to establish a cause-effect relationship. Recall bias and social desirability bias may have influenced the assessment of meal frequency, dietary diversity, and household food security. Unaddressed variables may have introduced residual confounders. Moreover, undernourished PWTB could not be linked to nutrition care centers due to the lack of nutritional supply for adult patients.

## Conclusion

This study revealed a high prevalence of undernutrition among adult PWTB. Anemia also presents severe public health significance in the population of interest in the Sidama Region. Dietary diversity score, food insecurity, TB-HIV coinfection,

and parasitic infection were predictors of undernutrition, while food insecurity, parasite infection, and TB-HIV coinfection predicted anemia among PWTB in the region.

## Recommendation

The synergistic effects of food insecurity, HIV coinfection, and parasitic infections underscore the need for integrated nutrition-TB care. We recommend routine nutritional screening and targeted interventions focused on food support, parasite control, and HIV management within TB treatment protocols to address this dual burden.

## Supporting information

**S1 Data. STATA Data Nutritional Status.**
(DTA)

## Acknowledgments

We would like to express our sincere gratitude to Yirgalem Hospital Medical College for granting us the opportunity to conduct this research. Our heartfelt thanks also go to the chief executive officers of the selected public hospitals for their invaluable support during data collection. We further extend our deep appreciation to the study participants, as well as the dedicated data collectors and supervisors, for their significant contributions to the success of this study.

## Author contributions

**Conceptualization:** Amelo Bolka, Selamawit Sikuare.

**Data curation:** Amelo Bolka, Assefa Philipos Kare.

**Formal analysis:** Amelo Bolka, Fentaw Wassie Feleke, Tafese Bosha.

**Funding acquisition:** Selamawit Sikuare.

**Investigation:** Selamawit Sikuare, Assefa Philipos Kare.

**Methodology:** Amelo Bolka, Tafese Bosha.

**Software:** Amelo Bolka, Fentaw Wassie Feleke.

**Supervision:** Assefa Philipos Kare, Tafese Bosha.

**Validation:** Assefa Philipos Kare, Fentaw Wassie Feleke.

**Writing – original draft:** Amelo Bolka, Selamawit Sikuare.

**Writing – review & editing:** Amelo Bolka, Selamawit Sikuare, Assefa Philipos Kare, Fentaw Wassie Feleke, Tafese Bosha.

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
