## [Decision Letter · Decision Letter 0]

9 Nov 2025

PGPH-D-25-02918

Nutritional Status and Associated Factors among Adult Tuberculosis Patients in Public Hospitals of Sidama Region, Ethiopia

Dear Dr. Bolka,

Thank you for submitting your manuscript to PLOS Global Public Health. After careful consideration, we feel that it has merit but does not fully meet PLOS Global Public Health’s publication criteria as it currently stands. Therefore, we invite you to submit a revised version of the manuscript that addresses the points raised during the review process.

You will see that Reviewer One has requested a modification to your methodological approach to be more theoretically grounded than you current, data-driven approach to selecting variables for inclusion in your multivariable regression.  I think that this is a valuable suggestion, and I hope that it will be possible for you to enhance your theory and therefore strengthen your paper.

We look forward to receiving your revised manuscript.

Kind regards,

Abraham D. Flaxman, Ph.D.

Academic Editor

Journal Requirements:

1. Please amend your online Financial Disclosure statement. If you did not receive any funding for this study, please simply state: “The authors received no specific funding for this work.”

2. Please update your online Competing Interests statement. If you have no competing interests to declare, please state: “The authors have declared that no competing interests exist.”

3. We note that your Data Availability Statement is currently as follows: “The data used in this study is freely available to any interested party and can be accessed without restriction”

Please confirm at this time whether or not your submission contains all raw data required to replicate the results of your study. Authors must share the “minimal data set” for their submission. PLOS defines the minimal data set to consist of the data required to replicate all study findings reported in the article, as well as related metadata and methods (https://journals.plos.org/globalpublichealth/s/data-availability#loc-minimal-data-set-definition).

If your submission does not contain these data, please either upload them as Supporting Information files or deposit them to a stable, public repository and provide us with the relevant URLs, DOIs, or accession numbers. For a list of recommended repositories, please see https://journals.plos.org/globalpublichealth/s/recommended-repositories.

4. Please include a separate legend or caption for each figure in your manuscript.

5. We have noticed that you have uploaded Supporting Information files (“S4_STATA Data_Nutritional Status.dta”), but you have not included a list of legends. Please add a full list of legends for your Supporting Information files before or after the references list.

Reviewers' comments:

Reviewer's Responses to Questions

**Comments to the Author**

1. Does this manuscript meet PLOS Global Public Health’s publication criteria?

Reviewer #1: Yes

Reviewer #2: Yes

2. Has the statistical analysis been performed appropriately and rigorously?

Reviewer #1: No

Reviewer #2: Yes

3. Have the authors made all data underlying the findings in their manuscript fully available (please refer to the Data Availability Statement at the start of the manuscript PDF file)?

Reviewer #1: Yes

Reviewer #2: Yes

4. Is the manuscript presented in an intelligible fashion and written in standard English?

Reviewer #1: Yes

Reviewer #2: Yes

Reviewer #1: This is an interesting study describing the prevalence of undernutrition, anaemia, parasitic infection and other measures of nutritional status among people with TB.

MAJOR COMMENTS

- The modelling approach is inconsistent with the aim of the study, which was to identify determinants of undernutrition and anaemia. Unless the overall aim was to derive a risk prediction algorithm, it is not appropriate to simply put all variables with an arbitrary p<0.25 in univariable analysis into a single multivariable model and see which ones come out as being associated. Please see here for further discussion on this topic: https://doi.org/10.1093/ije/dyaa213. I would suggest the authors redo the analysis using a DAG to inform the choice of confounders. We should be trying to move away from Table 2 approaches in our research: https://pubmed.ncbi.nlm.nih.gov/23371353/

MINOR COMMENTS

INTRODUCTION

- Should be updated to reflect new WHO guidelines (https://www.who.int/publications/i/item/9789240111967) and more recent evidence on BMI and TB (https://doi.org/10.1093/ije/dyaf154)

- Last paragraph does not effectively justify study rationale. The phrase "This gap underscores the need for further investigation to thoroughly understand the comprehensive effectiveness of the TB treatment program" is not relevant to the current manuscript. The aims need to be made more explicitly clear - estimate the prevalence of undernutrition and anaemia, and identify variables associated with these.

METHODS

- More detail is required on the inclusion/exclusion criteria, sampling methods, and participants. Presumably these were people with TB being treated as outpatients? Were they all recently diagnosed, or was it a cross section of anyone on TB treatment? I see in the results this is presented (87% newly diagnosed), but needs to be much clearer and more justified in the methods (e.g. recruiting people already on treatment will affect results, for various reasons).

- Please provide more detail on the way HFIAS is used to defined food secure or not, or add a reference.

- What justification is there for the arbitrary higher knowledge score of 13/16 rather than using this as a continuous variable?

- Only DS-TB, or also DR-TB?

- The statistical methods section needs more detail.

RESULTS

- Should be made more readable. There is no need to write out in full text numbers like "three hundred thirty-three". The narrative should be brief, with references made to the tables.

- For the wealthy index I would suggest lower, medium, or higher rather than poor, medium, or rich

- Severe, moderate, and mild undernutrition are not defined.

DISCUSSION

- I think should focus more on what the implications of these findings are for practice. At the moment it mostly contextualises against related findings from other settings.

Reviewer #2: This is a thoughtful and well-executed paper that addresses an important and underexplored intersection between tuberculosis, nutrition, and anemia. The authors have done an excellent job presenting clear, well-organized data using appropriate analytic methods. The inclusion of factors like dietary diversity, food insecurity, and parasitic infections adds real value to the literature, especially from a programmatic perspective. Given that there is dearth of data on the prevalence of malnutrition among persons with TB, particularly in Africa, this paper is an important addition.

A few small suggestions could help strengthen the manuscript even further:

Person-centered language

I’d recommend replacing the term TB patient with person with tuberculosis (PWTB) throughout. This aligns with WHO guidance and helps convey a more person-centered and respectful tone and one expected by TB civil society.

Clarifying the definition of undernutrition

The study defines undernutrition as BMI <18.5 kg/m², but it would be helpful to state explicitly that this operational definition corresponds to underweight. Providing additional granularity—such as severe (<16), moderate (16–16.9), and mild (17–18.4)—would parallel your nuanced classification of anemia and make the findings more interpretable. These cutoffs correspond to those recommended by the WHO

Recognizing hidden forms of malnutrition

It might strengthen the discussion to note that BMI alone may underestimate the true burden of nutritional deficiency. Many individuals with “normal” BMI likely have unmeasured micronutrient deficiencies, particularly given the low dietary diversity you document. A brief mention of this limitation would add nuance and align your discussion with the broader literature on “hidden hunger.”

**Do you want your identity to be public for this peer review?** For information about this choice, including consent withdrawal, please see our Privacy Policy

Reviewer #1: No

Reviewer #2: No

---

## [Decision Letter · Decision Letter 1]

10 Dec 2025

Nutritional Status and Associated Factors among Adult Patients with Tuberculosis  in Public Hospitals of Sidama Region, Ethiopia

PGPH-D-25-02918R1

Dear Mr. Bolka,

We are pleased to inform you that your manuscript 'Nutritional Status and Associated Factors among Adult Patients with Tuberculosis  in Public Hospitals of Sidama Region, Ethiopia' has been provisionally accepted for publication in PLOS Global Public Health.

Best regards,

Abraham D. Flaxman, Ph.D.

Academic Editor

Reviewer Comments (if any, and for reference):

Reviewer's Responses to Questions

**Comments to the Author**

Reviewer #1: All comments have been addressed

publication criteria?

Reviewer #1: Yes

3. Has the statistical analysis been performed appropriately and rigorously?

Reviewer #1: Yes

4. Have the authors made all data underlying the findings in their manuscript fully available (please refer to the Data Availability Statement at the start of the manuscript PDF file)?

Reviewer #1: Yes

5. Is the manuscript presented in an intelligible fashion and written in standard English?

Reviewer #1: No

Reviewer #1: (No Response)

**Do you want your identity to be public for this peer review?** For information about this choice, including consent withdrawal, please see our Privacy Policy

Reviewer #1: No
